# On the Active Adsorption of Chromium(III) from Alkaline Solutions Using Multiwalled Carbon Nanotubes

**Francisco José Alguacil** and **Félix A. López** *

National Center for Metallurgical Research (CENIM), Spanish National Research Council (CSIC), Avda. Gregorio del Amo, 8, 28040 Madrid, Spain; fjalgua@cenim.csic.es

* Correspondence: f.lopez@csic.es



**Featured Application: The paper constitutes a contribution and an advancement in the treatment of wastewater contaminated by chromium. These waters are present in various sectors of the metallurgical industry (pickling liquors and effluents from electroplating processes, among others).**

**Abstract:** The present investigation deals with the adsorption of chromium(III) from alkaline media, as representative of highly-caustic component solutions of nuclear tank wastes, using multiwalled carbon nanotubes. The adsorption of Cr(III) has been studied under various experimental conditions, i.e., stirring speed of the aqueous solution, initial metal and adsorbent concentrations, NaOH concentration in the aqueous solution, and temperature. The rate law indicated that chromium adsorption is well represented by the particle diffusion model, whereas the adsorption process fits with the pseudo-second order kinetic model within an exothermic setting. Equilibrium data fit to the Langmuir type-2 equilibrium isotherm in a spontaneous process. Chromium(III) can be eluted from metal-loaded nanotubes using acidic solutions, from which fine chromium(III) oxide pigment can ultimately be yielded.

**Keywords:** carbon nanotubes; chromium(III); adsorption; kinetics; alkaline medium

## 1. Introduction

Nanotechnologies are among the most important topics in today's investigations; among them, adsorptive nanomaterials have shown their tremendous potential, and different compositions and configurations of these nanomaterials are being investigated for different applications [1–6]. These techniques have been extensively used in separation science, i.e., metal recovery from pregnant and waste solutions, the obtaining of precious and strategic metals, and the treatment of effluents which include, e.g., toxic metals.

Included in the above, carbon nanomaterials in single-walled, multiwalled, and functionalized configurations are gaining considerable importance due to their application in the recovery and separation of metals from aqueous phases [7–12].

Chromium is a toxic element [13], especially in its VI oxidation state, but solutions containing chromium(III) are also considered to be hazardous in nature due to the real potential of oxidation to the VI state; thus, its elimination from different aqueous solutions should be considered a major priority. The recovery and separation of chromium(III) from aqueous solutions with different materials and nanomaterials have been reported in the literature, with about 60 papers dealing with the matter in 2019 [14]; however, the majority of this information is related to the removal of $Cr^{3+}$ from acidic to

neutral aqueous pH values, and only scarce information is available on the use of carbon nanotubes as adsorbents for Cr(III) separation and/or speciation in an alkaline medium [15,16].

In the present work, the Cr(III) adsorption results are presented from alkaline aqueous solutions using multiwalled carbon nanotubes. This alkaline medium is representative of the solutions which can be encountered or dumped in nuclear tank wastes. Different experimental variables affecting Cr(III) uptake were investigated, together with the fit of experimental data to various models. Finally, Cr(III) desorption is accomplished by the use of acidic solutions.

## 2. Experimental

Multiwalled carbon nanotubes (MWCNTs) were obtained from Fluka, and were used, unless otherwise stated, without further modification. Their main characteristics are shown in Table 1. The Z potential was estimated as indicated in the literature [17].

All chemicals were of AR grade and used in the experiments directly without further purification. Other adsorbents used in this work were also obtained from Fluka or Sigma-Aldrich (oxidized multiwalled carbon nanotubes (ox-MWCNTs), Ionac SR7, Lewatit EP63), except for the activated carbon, which was generated as described in the literature [18]. Throughout the experimentation, distilled water was used.

All experiments were conducted in a glass reactor at 20 °C, except for those performed at various temperatures, by using batch technique. Once the aqueous solution and the carbon nanotubes were put into the reactor and mixed using a four-blade glass impeller (25 mm diameter), aliquots were taken at given times in order to analyze the chromium content in the solution by AAS. Metal uptake onto the nanotubes was calculated by the mass balance.

**Table 1.** Characteristics of the multiwalled carbon nanotubes.

| Type | Multiwalled |
|---|---|
| melting range | 3652–3697 °C |
| density | 2.1 g mL$^{-1}$ |
| appearance | dust |
| purity | 98% carbon |
| dimensions | 10 ± 1 nm external diameter |
| | 4.5 ± 0.5 nm internal diameter |
| | 3–6 μm (length) |
| maximum adsorption | 1295 cm$^3$ g$^{-1}$ |
| BET | 263 m$^2$ g$^{-1}$ |
| Z potential | 1.22 |

### 2.1. Modeling the Rate Law

Experimental results were best fitted ($r^2$ = 0.962) to the particle-diffusion controlled process, i.e., the rate equation represented as:

$$\ln\left(1 - F^2\right) = -kt \tag{1}$$

In the above expression, F is the factorial approach to the equilibrium, defined as:

$$F = \frac{[Cr(III)]_{cnt,t}}{[Cr(III)]_{cnt,e}} \tag{2}$$

being $[Cr(III)]_{cnt,t}$ and $[Cr(III)]_{cnt,e}$ chromium(III) concentrations in the carbon nanotubes at an elapsed time and at equilibrium, respectively.

### 2.2. Modeling of Kinetic Adsorption

In this case, the experimental data best fit to the pseudo-second-order kinetic model

$$\frac{t}{[Cr(III)]_{cnt,t}} = \frac{1}{k_2[Cr(III)]_{cnt,e}^2} + \frac{1}{[Cr(III)]_{cnt,e}}t \tag{3}$$

### 2.3. Modeling of Adsorption Isotherm

The results obtained from this work fit well ($r^2 = 0.921$) with the Langmuir Type-2 isotherm, represented in its linear form as:

$$\frac{1}{[Cr(III)]_{cnt,e}} = \frac{1}{[Cr(III)]_{cnt,m}} + \frac{1}{k_L[Cr(III)]_{cnt,m}} \frac{1}{[Cr(III)]_{aq,e}} \tag{4}$$

where $k_L$ is the Langmuir constant, $[Cr(III)]_{cnt,m}$ the maximum metal uptake onto the nanotubes, and $[Cr(III)]_{cnt,e}$ and $[Cr(III)]_{aq,e}$ the metal equilibrium concentrations in the carbon nanotubes and in the aqueous solution, respectively.

## 3. Results and Discussion

### 3.1. Influence of Stirring Speed

The influence of stirring speed was studied in order to achieve a uniform mixture of the aqueous solution and the adsorbent, and to minimize the thickness of the aqueous boundary layer, with the aqueous solution and adsorbent conditions being maintained as follows: Cr(III) 0.01 g L$^{-1}$ in 0.1 M NaOH, and adsorbent dosage of 1g L$^{-1}$.

The results derived from this investigation revealed that the stirring speed had no influence on the time required for the system to reach equilibrium (Figure 1), since for every stirring speed investigated here, at 30 min of contact time between the aqueous solution and the adsorbent, 85% of the chromium(III) from the solution had been adsorbed onto the carbon nanotubes, whereas equilibrium was reached after 120 min at all stirring speeds. However, the stirring speed had an influence on the maximum metal uptake onto the nanotubes (Table 2). As shown from these results, a maximum chromium(III) uptake was achieved at 1000–1500 min$^{-1}$, with this being attributable to the fact that at these stirring speeds, the minimum of the aqueous layer was reached and adsorption was maximized. A stirring speed of 1000 min$^{-1}$ was kept constant throughout the experiments.

The data at 1000 min$^{-1}$ were used to estimate the rate law in the present system. The experimental results fit best with the particle diffusion model, as represented by Equation (1), with a rate constant of $3.8 \times 10^{-2}$ min$^{-1}$.

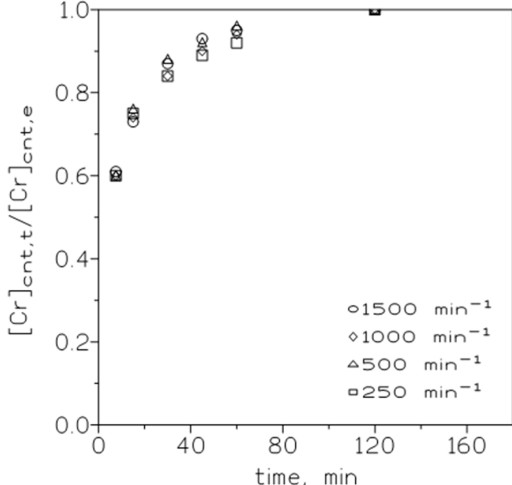

**Figure 1.** Dimensionless [Cr(III)]cnt,t/[Cr(III)]cnt,e versus time at various stirring speeds. ([Cr(III)]cnt,t and [Cr(III)]cnt,e are the chromium concentrations in the nanotubes at elapsed time and at equilibrium, respectively).

**Table 2.** Chromium(III) adsorption at various stirring speeds (Temperature: 20 °C. Time: 2 h).

| Speed (min$^{-1}$) | Cr(III) Uptake (mg g$^{-1}$ Adsorption) |
|---|---|
| 250 | 6.4 |
| 500 | 7.0 |
| 1000 | 8.4 |
| 1500 | 8.3 |

*3.2. Influence of the NaOH Concentration in the Aqueous Solution*

Previous experiments have shown that from Cr(III)-bearing alkaline solutions, i.e., 0.01 g L$^{-1}$ Cr(III) in 0.1 M NaOH, there was no Cr(III) removal-precipitation over a period of 3 h and under agitation at 1000 min$^{-1}$ in the absence of the carbon nanotubes; this indicated that the removal of Cr(III) from these alkaline solutions was solely due to the presence of carbon nanotubes acting as an adsorbent material of the heavy metal. In order to assess the influence of this variable during the adsorption of chromium(III), NaOH concentration variations in the range 0.1–0.5 M NaOH were carried out with aqueous solutions containing 0.01 g L$^{-1}$ Cr(III). The adsorbent dosage was 1 g L$^{-1}$ and the temperature was 20 °C. The results from this experiment (Table 3) showed that the percentage of metal adsorption (and metal uptake) was dependent of the NaOH concentration of the aqueous solution, increasing the chromium(III) loaded onto the adsorbent from 0.1 to 0.35 M NaOH solutions, and decreasing as the NaOH concentration in the solution was further increased.

**Table 3.** Influence of the NaOH concentration on chromium(III) adsorption onto the nanotubes.

| NaOH, M | Cr(III) Adsorption (%) | Metal Uptake (mg g$^{-1}$) |
|---|---|---|
| 0.1 | 84 | 8.4 |
| 0.2 | 89 | 8.9 |
| 0.28 | 96 | 9.6 |
| 0.35 | 99 | 9.9 |
| 0.43 | 51 | 5.1 |
| 0.5 | 37 | 3.7 |

*3.3. Influence of Adsorbent Dosage on the Adsorption of Chromium(III)*

The results concerning the adsorption of chromium(III) from the aqueous solution containing this metal in a 0.1 M NaOH medium and various adsorbent doses in the range 0.25–4 g L$^{-1}$ revealed an increase in the percentage of adsorption as the adsorbent dosage was increased (Figure 2).

The experimental results of the chromium(III) adsorption onto the nanotubes were fit to various kinetics models; from these fits, it was assumed that the pseudo-second order kinetic model, Equation (3), best represented the adsorption kinetics for the initial carbon nanotubes dosages in the 0.25–4 g L$^{-1}$ range.

These results confirmed that a chemisorption process conveniently described the measured Cr(III) removal from the aqueous solution. Table 4 summarizes the results from this fit.

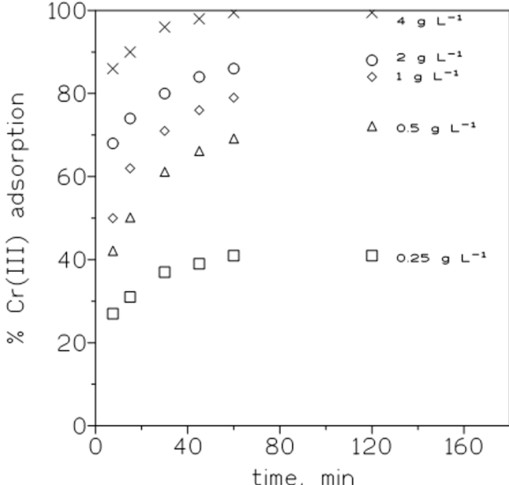

**Figure 2.** Influence of adsorbent dosage on chromium(III) adsorption. Aqueous solution: 0.01 g L$^{-1}$ Cr(III) in 0.1 M NaOH. Temperature: 20 °C.

**Table 4.** Parameters for the fit of Equation (3) to various nanotubes doses.

| Nanotubes Dosage (g L$^{-1}$) | $r^2$ | $k_2$, (g min$^{-1}$ mg$^{-1}$) |
|:---:|:---:|:---:|
| 0.25 | 0.9986 | $1.8 \times 10^{-1}$ |
| 1 | 0.9997 | $1.8 \times 10^{-2}$ |
| 4 | 0.9998 | $1.4 \times 10^{-2}$ |

The equilibrium data fitted to the Langmuir Type-2 isotherm model, Equation (4), and the corresponding plot in Figure 3. The relative parameters for this fit are [Cr(III)]$_{cnt,m}$ = 204 mg g$^{-1}$ and $k_L$ = 2.3 × 10$^{-2}$ L mg$^{-1}$. Moreover, by using the next expression,

$$R = \frac{1}{1 + k_L [Cr(III)]_{aq,0}^2} \tag{5}$$

where [Cr(III)]$_{aq,0}$ is the initial metal concentration in the solution (10 mg L$^{-1}$), it is assumed that the adsorption process is favorable (R = 0.31), since 0 < R < 1.

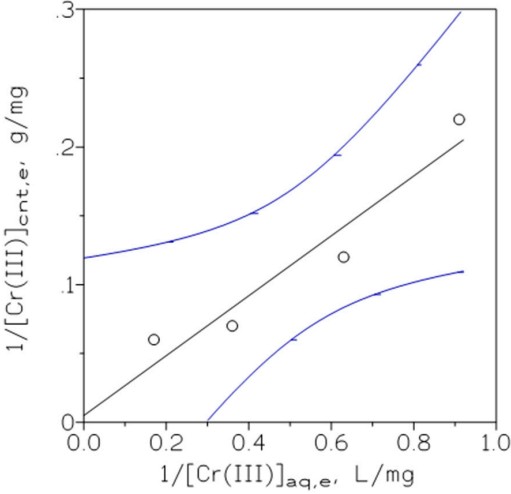

**Figure 3.** Plot of 1/[Cr(III)]$_{cnt,e}$ *versus* 1/[Cr(III)]$_{aq,e}$ for the Langmuir Type-2 equation. Experimental conditions are as in Figure 2. Time: 2 h. Dotted lines represented 95% confidence interval of the regression line.

### 3.4. Influence of Temperature on the Adsorption of Chromium(III)

The adsorption of chromium(III) (0.01 g $L^{-1}$) in 0.1 M NaOH by 1 g $L^{-1}$ of the adsorbent was studied over a temperature range of 20–60 °C. Under these conditions, increasing temperature gave a decrease in chromium adsorption (84% at 20 °C versus 52% at 60 °C), and defined the metal distribution coefficient between the adsorbent and the aqueous solution as:

$$D_{Cr} = \frac{[Cr(III)]_{cnt,e}}{[Cr(III)]_{aq,0}} \tag{6}$$

The plot of the log $D_{Cr}$ versus 1000/T was linear ($r^2$ = 0.9959) (Figure 4). From this plot, the value of ΔH° was estimated to be −32 kJ $mol^{-1}$; thus, the adsorption was exothermic, and $\Delta S^{o0}$ = −95 J $mol^{-1}$ $K^{-1}$, indicating a decrease in the randomness at the solid-liquid interface during the adsorption process. The estimated $\Delta G^0$ value was −4 kJ $mol^{-1}$, which represented a spontaneous adsorption process.

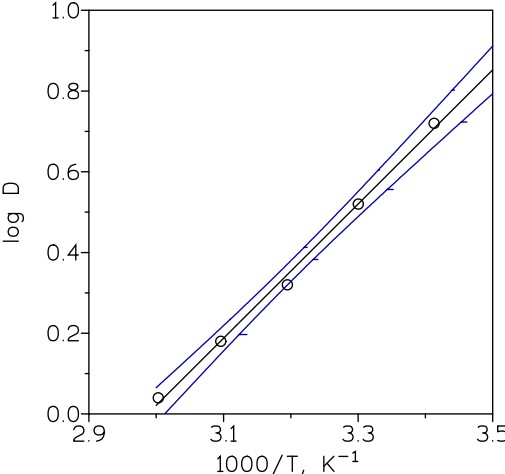

**Figure 4.** Plot of log D versus 1000/T for the adsorption of chromium(III) at various temperatures. Dotted lines represented a 95% confidence interval of the regression line.

### 3.5. Chromium(III) Removal from the Aqueous Solution Using Different Adsorbents/Ion Exchangers: A Comparison

Different adsorbents or anion exchange resins were used to compare chromium(III) adsorption results with that obtained with the present multiwalled carbon nanotubes. In the present case, the aqueous solution was of 0.01 g $L^{-1}$ Cr(III) at 0.1 M NaOH, with an adsorbent dosage of 1 g $L^{-1}$. The temperature of the experiments was 20 °C. The results from this comparison are summarized in Table 5. As shown, the worst results were obtained when the nonfunctionalized Lewatit EP63 resin was used to remove Cr(III) from the solution, whereas the use of active carbon achieved near 94% Cr(III) removal value, which was very near the yield reached with the use of the multiwalled carbon nanotubes.

**Table 5.** Chromium(III) adsorption using various adsorbents.

| Adsorbent | Functional Group | [a] Cr(III) (mg $g^{-1}$) |
|---|---|---|
| MWCNTs | none | 8.4 |
| Ionac SR7 | [b] QAS | 7.3 |
| Lewatit EP63 | none | 1.1 |
| ox-MWCNTs | carboxylic | 5.0 |
| Active carbon [19,20] | none | 9.4 |

[a] At equilibrium. [b] Quaternary ammonium salt. Ionac SR7 and Lewatit EP63 are resins.

*3.6. Desorption*

Previous experiments demonstrated that at pH 4, a mere 5% of the chromium(III) was adsorbed by these nanotubes, thus it seemed logical to approach the desorption process through the use of acidic solutions. The results obtained when 8.4 mg g$^{-1}$ Cr(III)-loaded carbon nanotubes were put into contact with acidic solutions allowed us to conclude that through the use of sulfuric acid solutions (0.1 M onwards), nearly 85% of the chromium(III) loaded onto the nanotubes could be desorbed after 30 min of reaction at 20 °C. From these solutions, a greenish, fine pigment was yielded [21].

## 4. Conclusions

The developed investigation was useful for the recovery of chromium(III) from alkaline conditions, although the adsorption was dependent upon the NaOH concentration in the aqueous solution. The adsorption of Cr(III) onto multiwalled carbon nanotubes was quick at the initial contact times but slowed down with increasing contact times. The rate law was well described by the particle diffusion model in an exothermic ($\Delta H^{\circ} = -32$ kJ mol$^{-1}$) adsorption process. In the range of carbon nanotubes dosages (0.25–4 g L$^{-1}$) used in the investigation, adsorptions can be described by the pseudo-second order kinetic model, whereas the Langmuir type-2 isotherm described well the Cr(III) adsorption in a spontaneous process. The multiwalled carbon nanotubes used in the investigation performed well in terms of chromium(III) removal from an alkaline medium in comparison to a number of other Cr(III)-potential adsorbents. Acidic solutions can be used to desorb chromium(III) from metal-loaded nanotubes, and Cr(III) can be recovered as a fine pigment. The results derived from this investigation represent a necessary contribution to the understanding of heavy metal adsorption on nanomaterials from caustic environments, such as complex radioactive liquid wastes.

**Author Contributions:** F.A.L. Funding acquisition; F.J.A. and F.A.L. methodology; F.J.A. formal analysis; F.J.A. and F.A.L. investigation; F.J.A. writing—original draft; F.J.A. and F.A.L. writing—review & editing. All authors have read and agreed to the published version of the manuscript.

**Funding:** This research received no external funding.

**Acknowledgments:** We acknowledge support of the publication fee by the CSIC Open Access Publication Support Initiative through its Unit of Information Resources for Research (URICI). The authors appreciate Martin Ian Maher (CENIM-CSIC) collaboration in the grammar correction of the manuscript.

**Conflicts of Interest:** The authors declare no conflict of interest.

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
