# Peer review of "On the Active Adsorption of Chromium(III) from Alkaline Solutions Using Multiwalled Carbon Nanotubes"

_applsci, doi:10.3390/app10010036_

Round 1

Reviewer 1 Report

Superficially written manuscript.

This kind of research and manuscripts are usually provide much more informations on the adsorption capacity of the examined adsorbent, especially about correlations among adsorbent efficiency and other process parameters.

Therefore, before acceptance of this manuscript, I suggest to authors to provide more information about adsorbent efficiency and extent the manuscript.

Author Response

Reviewer 1.

Superficially written manuscript.

This kind of research and manuscripts are usually provide much more information’s on the adsorption capacity of the examined adsorbent, especially about correlations among adsorbent efficiency and other process parameters.

Therefore, before acceptance of this manuscript, I suggest to authors to provide more information about adsorbent efficiency and extent the manuscript.

We appreciate the comments by the reviewer, but regrettably we can not agree with them, since and based in our experience of more than 30 years in the scientific business,  we know how to write a manuscript, and if you compare the results presented here with other presented in adsorption papers,  one can see that our manuscript give enough information to understand the adsorption-desorption process, the latter not always included in this type of investigations. Moreover, we included data about one experimental parameter: the influence of the stirring speed on metal adsorption, that was usually neglected by many authors.

In any case, we performed some modifications on the revised manuscript.   

Reviewer 2 Report

The manuscript reports the investigation of Cr(III) adsorption on commercially available multi-walled carbon nanotubes. However, it lacks novelty and serious experimental flaws were observed. Some comments are listed as follows:

The English level of this manuscript should be extensively improved. Insufficient introduction (state of the art, present study description etc). Although the authors claim as application the wastewater treatment, the experimental procedure was carried out only in distilled water. As it is known, Cr(III) has extremely low solubility in aqueous solutions. Therefore, the use of high concentrations (10 mg/L) in combination with the application of high alkaline conditions, metal’s precipitation is favored (maybe on the adsorbent). So, a better explanation of the removal’s mechanism should be given, taking into account the above, possibly by repeating Table 3 without the use of the adsorbent (blank). In section 3.3, Table 4 presents some of the experimental parameters, according to Figure 2. Also, regarding Langmuir model, a clarification should be added to the equilibrium conditions mentioned (especially for time) and the addition of the corresponding figure. In section 3.4, authors should provide the mentioned plot in a figure and a comparison of their results with the literature. The authors present and explain the applied equations in “Results and discussion” section, while they are part of the methodology, so they should be mentioned in “Materials and Methods”.

Author Response

Reviewer 2.

We appreciated your comments, please see below our response to them.

The manuscript reports the investigation of Cr(III) adsorption on commercially available multi-walled carbon nanotubes. However, it lacks novelty and serious experimental flaws were observed. Some comments are listed as follows.

We can not be agree with the comment, precisely the novelty of the manuscript is the investigation of Cr(III) removal from alkaline solutions, with this nanotubes, and the other adsorbents, and not the usual acidic or almost neutral (pH 6) conditions used in works related with the removal of this element. 

The English level of this manuscript should be extensively improved.

Again we can not agree with this comment, after publishing many many papers in SCI Journals, it is out of discussion that we know how to write papers.

Insufficient introduction (state of the art, present study description etc).

Again we disagree with this comment, we are also reviewers for many Journals, and in a number of cases authors longer the introduction too much and introducing superfluous information. We prefer to be concise about the manuscript overview, please don not forget that it is our manuscript.

Although the authors claim as application the wastewater treatment, the experimental procedure was carried out only in distilled water.

Please note that we only used this word “wastewater” as a keyword, we did not use it anymore along the manuscript, we deleted the word in the revised version of the manuscript. In this type of investigation, it is a very normal occurrence to use distilled water, MiliQ water, etc., to dissolved the compound containing the metal which adsorption is going to be investigated. The rare situation is the use of real solutions.

As it is known, Cr(III) has extremely low solubility in aqueous solutions. Therefore, the use of high concentrations (10 mg/L) in combination with the application of high alkaline conditions, precipitation is favored (maybe on the adsorbent). So, a better explanation of the removal mechanism should be given, taking into account the above, possibly by repeating Table 3 without the use of the adsorbent (blank).

We both know the behavior of Cr(III) in solutions, please note that we both are Ph.D. in Chemistry since a long time (more than 30 years). Anyway, please note that we add a new paragraph in subsection 3 (Influence of NaOH concentration on Cr(III) adsorption). There are not precipitation at 0.1 M NaOH; and in relation with the high concentration (reviewer comment: 10 mg/L) used in this work, we can not be agree, since the level of concentrations appearing in effluents vary from the traces to the g/L order; in fact, these authors are currently working with an industrial waste solution in which the level of the impurity is the same than that of the valuable metal (about 60 g/L each).

In section 3.3, Table 4 presents some of the experimental parameters, according to Figure 2. Also, regarding Langmuir model, a clarification should be added to the equilibrium conditions mentioned (especially for time) and the addition of the corresponding figure.

We did the above: inclusion of a figure (Figure 3 in the revised manuscript).

In section 3.4, authors should provide the mentioned plot in a figure and a comparison of their results with the literature.

We included the suggested plot (Figure 4 in the revised manuscript). Please note that comparison is not possible since the experimental conditions are not the same, i.e. acidic versus alkaline in this case.

The authors present and explain the applied equations in Results and discussion section, while they are part of the methodology, so they should be mentioned in Materials and Methods.

For us it is a surprise that the reviewer mentioned the above, because in most, if not all, of the papers consulted by us, and not only in the case of Cr(III), these equations are put in the convenient subsection and not in Experimental or Materials and Methods sections. However, to be constructive we rewrite the manuscript putting the equations in Experimental section, and then mentioned them in their precise site.

Round 2

Reviewer 1 Report

Professional text editing (English language and style are fine/minor spell check required) is recommended prior publication.

Author Response

Regarding the writing in English, we have submitted the work to a native researcher and his conclusion is that the manuscript is grammatically correct. Of course it could be written differently but the manuscript can be perfectly understood by any reader of the journal, even if he is not an expert in the field.

Reviewer 2 Report

Initially, I would like to make it clear that all my previous comments were aimed at improving the manuscript in order to increase the interest of other researchers towards it. In my opinion, clarifying certain points and adding data (which existed) actually helped to the previously mentioned statement. On the level of English, my disagreement concerns the use of certain expressions and we can only conclude that it is a grey zone in general.

Author Response

We welcome the additional comments of the reviewer. Our goal is also to try to make our research of interest to the scientific community. We appreciate the efforts of the reviewers (we are too). They are fundamental in the process of dissemination of research results.
Regarding the writing in English, we have submitted the work to a native researcher and his conclusion is that the manuscript is grammatically correct. Of course it could be written differently but the manuscript can be perfectly understood by any reader of the journal, even if he is not an expert in the field.